# A Lazy Approach to Long-Horizon Gradient-Based Meta-Learning

## Abstract

Gradient-based meta-learning relates task-specific models to a meta-model by gradients. By this design, an algorithm first optimizes the task-specific models by an inner loop and then backpropagates meta-gradients through the loop to update the meta-model. The number of inner-loop optimization steps has to be small (e.g., one step) to avoid high-order derivatives, big memory footprints, and the risk of vanishing or exploding meta-gradients. We propose an intuitive teacher-student scheme to enable the gradient-based meta-learning algorithms to explore long horizons by the inner loop. The key idea is to employ a student network to adequately explore the search space of task-specific models (e.g., by more than ten steps), and a teacher then takes a "leap" toward the regions probed by the student. The teacher not only arrives at a high-quality model but also defines a lightweight computation graph for meta-gradients. Our approach is generic; it performs well when applied to four meta-learning algorithms over three tasks: few-shot learning, long-tailed classification, and meta-attack.

## 1 Introduction

Humans can quickly learn the skills needed for new tasks by drawing from a fund of prior knowledge and experience. To grant machine learners this level of intelligence, meta-learning studies how to leverage past learning experiences to more efficiently learn for a new task (Vilalta & Drissi, 2002). A hallmark experiment design provides a meta-learner a variety of few-shot learning tasks (meta-training) and then desires it to solve previously unseen and yet related few-shot learning tasks (meta-test). This design enforces "learning to learn" because the few-shot training examples are insufficient for a learner to achieve high accuracy on any task in isolation.

Recent meta-learning methods hinge on deep neural networks. Some work learns a recurrent neural network as an update rule to a model (Ravi & Larochelle, 2016; Andrychowicz et al., 2016). Another line of methods transfers an attention scheme across tasks (Mishra et al., 2017; Vinyals et al., 2016a). Gradient-based meta-learning gains momenta recently following the seminal work (Finn et al., 2017). It is model-agnostic meta-learning (MAML), learning a global model initialization from which a meta-learner can quickly derive task-specific models by using a few training examples.

In its core, MAML is a bilevel optimization problem (Colson et al., 2007). The upper level searches for the best global initialization, and the lower level optimizes individual models, which all share the common initialization, for particular tasks sampled from a task distribution. This problem is hard to solve. Finn et al. (2017) instead propose a "greedy" algorithm, which comprises two loops. The inner loop samples tasks and updates the task-specific models by $k$ steps using the tasks' training examples. The $k$-step updates write a differentiable computation graph. The outer loop updates the common initialization by backpropagating meta-gradients through the computation graph. This method is "greedy" in that the number of inner steps is often small (e.g., $k = 1$). The outer loop takes actions before the inner loop sufficiently explores its search space.

This "greedy" algorithm is due to practical constraints that backpropagating meta-gradients through the inner loop incurs high-order derivatives, big memory footprints, and the risk of vanishing or exploding gradients. For the same reason, some related work also turns to greedy strategies, such as meta-attack (Du et al., 2019) and learning to reweigh examples (Ren et al., 2018b).

To this end, it is natural to pose at least two questions. *Would a less greedy gradient-based meta-learner (say, $k > 10$ inner-loop updates) achieve better performance? How to make it less greedy?*

To answer these questions, we provide some preliminary results by introducing a lookahead optimizer (Zhang et al., 2019) into the inner loop. It is intuitive to describe it as a teacher-student scheme. We use a student neural network to explore the search space for a given task adequately (by a large number $k$ of updates), and a teacher network then takes a "leap" toward the regions visited by the student. As a result, the teacher network not only arrives at a high-performing model but also defines a very lightweight computation graph for the outer loop. In contrast to the traditionally "greedy" meta-learning framework used in MAML (Finn et al., 2017), meta-attack (Du et al., 2019), learning to reweigh examples (Ren et al., 2018b), etc., our approach has a "lazy" teacher. It sends a student to optimize for a task up to many steps and moves only once after that.

Rajeswaran et al. (2019) proposed a less "greedy" MAML, with which this work shares a similar goal, but our approach improves the gradient-based meta-learning framework rather than a particular algorithm. Hence, we evaluate it on different methods and tasks, including MAML and Reptile (Nichol et al., 2018) for few-shot learning, a two-component weighting algorithm (Jamal et al., 2020) for long-tailed classification, and meta-attack (Du et al., 2019). Extensive results provide an affirmative answer to the first question above: long-horizon exploration in the inner loop improves a meta-learner's performance. We expect our approach, along with the compelling experimental results, can facilitate future work to address the second question above.

## 2 "GREEDY" GRADIENT-BASED META-LEARNING

We first review gradient-based meta-learning from the perspective of "search space carving".

**Notations.** Let $P_\mathcal{T}$ denote a task distribution. For each task drawn from the distribution $\mathcal{T} \sim P_\mathcal{T}$, we have a training set $\mathcal{D}_{tr}$ and a validation set $\mathcal{D}_{val}$, both in the form of $\{(x_1, y_1), (x_2, y_2), \cdots\}$ where $x_m$ and $y_m$ are respectively an input and a label. We learn a predictive model for the task by minimizing an empirical loss $\mathcal{L}^\mathcal{T}_{\mathcal{D}_{tr}}(\phi)$ (e.g., cross-entropy) over the training set while using the validation set to choose hyper-parameters (e.g., early stopping), where $\phi$ collects all trainable parameters of the model. Similarly, we denote by $\mathcal{L}^\mathcal{T}_{\mathcal{D}_{val}}(\phi)$ the loss calculated over the validation set.

**Meta-learning as "space carving".** Instead of focusing on an isolated task, meta-learning takes a global view and introduces a meta-model, parameterized by $\theta$, that can improve the learning efficiency for all individual tasks drawn from the task distribution $P_\mathcal{T}$. The underlying idea is to derive a task-specific model $\phi$ from not only the training set $\mathcal{D}_{tr}$ but also the meta-model $\theta$, i.e., $\phi \in \mathcal{M}(\theta, \mathcal{D}_{tr})$. We refer to $\mathcal{M}(\theta, \mathcal{D}_{tr})$ the "carved" search space for the task-specific model $\phi$, where the "carving" function is realized as an attention module in (Vinyals et al., 2016a; Mishra et al., 2017), as a conditional neural process in (Garnelo et al., 2018; Gordon et al., 2020), as a gradient-based update rule in (Finn et al., 2017; Park & Oliva, 2019; Li et al., 2017; Nichol et al., 2018), and as a regularized optimization problem in (Rajeswaran et al., 2019; Zhou et al., 2019).

An optimal meta-model $\theta^*$ is supposed to yield the best task-specific models in expectation,

$$\theta^* \leftarrow \arg\min_\theta \mathbb{E}_{\mathcal{T} \sim P_\mathcal{T}, \mathcal{D}_{val} \sim \mathcal{T}} \mathcal{L}^\mathcal{T}_{\mathcal{D}_{val}}(\phi^*(\theta)) \text{ subject to } \phi^*(\theta) \leftarrow \arg\min_{\phi \in \mathcal{M}(\theta, \mathcal{D}_{tr})} \mathcal{L}^\mathcal{T}_{\mathcal{D}_{tr}}(\phi). \quad (1)$$

One can estimate the optimal meta-model $\theta^*$ from some tasks and then use it to "carve" the search space, $\mathcal{M}(\theta^*, \mathcal{D}_{tr})$, for novel tasks' models.

**Gradient-based meta-learning.** One of the notable meta-learning methods is MAML (Finn et al., 2017), which uses a gradient-based update rule to "carve" the search space for a task-specific model,

$$\mathcal{M}_{\text{MAML}}(\theta, \mathcal{D}_{tr}) := \{\phi_0 \leftarrow \theta\} \cup \{\phi_j \,|\, \phi_j \leftarrow \phi_{j-1} - \alpha \nabla_\phi \mathcal{L}^\mathcal{T}_{\mathcal{D}_{tr}}(\phi_{j-1}), \ j = 1, 2, \cdots, k\} \quad (2)$$

where the meta-model $\theta$ becomes an initialization to the task-specific model $\phi_0$, the other candidate models $\phi_1, \cdots, \phi_k$ are obtained by gradient descent, and $\alpha > 0$ is a learning rate. Substituting it into equation (1), $\phi_k \in \mathcal{M}_{\text{MAML}}(\theta, \mathcal{D}_{tr})$ is naturally a solution to the lower-level optimization problem, and MAML solves the upper-level optimization problem by gradient descent,

$$\theta \leftarrow \theta - \beta \mathbb{E}_{\mathcal{T} \sim P_\mathcal{T}, \mathcal{D}_{val} \sim \mathcal{T}} \nabla_\theta \mathcal{L}^\mathcal{T}_{\mathcal{D}_{val}}(\phi_k(\theta)), \quad (3)$$

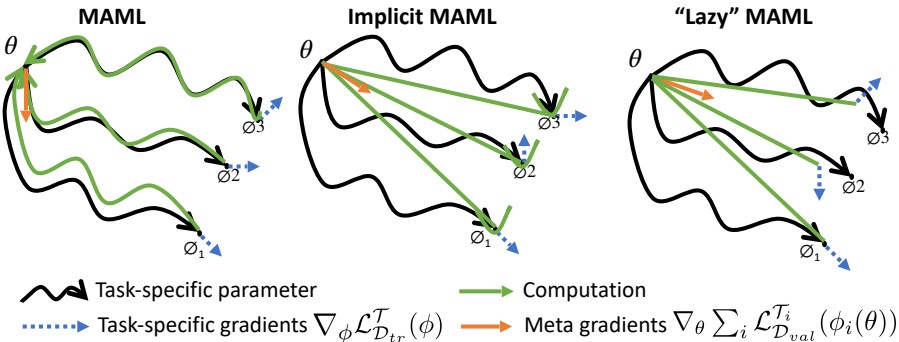

Figure 1: To compute the meta-gradients $\nabla_\theta \sum_i \mathcal{L}_{\mathcal{D}_{val}}^{\mathcal{T}_i}(\phi_i(\theta))$, MAML (Finn et al., 2017) differentiates through the inner updates, the implicit MAML (Rajeswaran et al., 2019) approximates local curvatures, while we differentiate through the "lazy" teacher's one-step "leap". The exploratory student may make many steps of inner updates before the teacher's "leap".

where $\beta$ is a learning rate, and $\phi_k(\theta)$ indicates the dependency on the meta-model $\theta$. The gradient must backpropagate through the chain of updates in eq. (2), which has to be short (e.g., $k = 1$) to avoid big memory footprints, high-order derivatives, and the risk of vanishing or exploding gradients.

We say MAML is "greedy" in that it descends meta-gradients for the meta-model $\theta$ before it runs adequate updates to the task-specific model $\phi$. As an increasing number of works adopt the gradient-based "search space carving" for task-specific models (Li et al., 2017; Rajeswaran et al., 2019; Park & Oliva, 2019; Flennerhag et al., 2019; Yin et al., 2019), they also bear greedy algorithms. Relaxing the greedy strategy may benefit not one, but a variety of, meta-learning methods and tasks.

## 3 A "LAZY" APPROACH TO GRADIENT-BASED META-LEARNING

In this section, we describe a "lazy" meta-learning approach, which is readily applicable to different gradient-based meta-learning algorithms. We first describe the general approach as an improvement to MAML and then customize it for few-shot learning, long-tailed classification, and meta-attack.

### 3.1 GENERAL APPROACH

Given a meta-model $\theta$, we "carve" the search space for task-specific models $\phi \in \mathcal{M}(\theta, \mathcal{D}_{tr})$ by a teacher-student scheme. The key idea is to let a student explore the search space adequately using the training set of a task-specific model without worrying the length of the update chain because a teacher will examine the explored regions by the student, followed by a one-step "leap". Hence, one can update the meta-model by backpropagating gradients through the teacher's "leap", not the student's update chain (ignoring that the chain starts from the meta-model). Figure 1 illustrates the main idea.

**An exploratory student** acts exactly the same as the gradient-based updates in MAML except that it explores the feasible space by a large number of steps ($k > 10$), resulting in $k + 1$ checkpoints of a task-specific model $\phi \in \mathcal{M}_{\text{MAML}}(\theta, \mathcal{D}_{tr}) = \{\phi_j, j = 0, \cdots, k\}$. It is clear from Section 2 that we cannot backpropagate the meta-gradients through the long chain of checkpoints, $\phi_0, \cdots, \phi_k$, made by the exploratory student.

**A lazy teacher** sits at the initialization $\phi_0 = \theta$ till the student stops. It then takes a "leap" towards the region explored by the student. The teacher essentially defines another "carved search space" for the task-specific model $\phi$,

$$\mathcal{M}_{\text{LAZY}}(\theta, \mathcal{D}_{tr}) := \gamma\theta + (1 - \gamma)\mathcal{R}_{k-b+1...k} \tag{4}$$

where $\gamma \in [0, 1]$. The region $\mathcal{R}_{k-b+1...k}$ is a convex hull of the last $b$ checkpoints the student visited:

$$\mathcal{R}_{k-b+1...k} := \alpha_{k-b+1}\phi_{k-b+1} + \alpha_{k-b+2}\phi_{k-b+2} + \cdots + \alpha_k\phi_k, \tag{5}$$

where the coefficients $\{\alpha\}$ are non-negative and their sum equals 1, i.e., $\alpha_{k-b+1} + \cdots + \alpha_k = 1$. The last $b$ checkpoints presumably cover a high-quality task-specific model $\phi$ by a better chance than the first few checkpoints. We shall experiment with $b = 3$ and $b = 1$.

Any task-specific model $\phi$ in this "lazy" space $\mathcal{M}_{\text{LAZY}}(\theta, \mathcal{D}_{tr})$ is determined by the hyper-parameters $\gamma$ and $\alpha_{k-b+1}, \cdots, \alpha_k$, over which we conduct a grid search to minimize the validation loss $\mathcal{L}^{\mathcal{T}}_{\mathcal{D}_{val}}(\phi)$. This is similar in spirit to meta-SGD (Li et al., 2017), which uses the validation data to search for the hyper-parameter of learning rates.

Denote by $\hat{\gamma}\theta + (1 - \hat{\gamma})\hat{\phi}$ the task-specific model as a result of the grid search. Notably, it is only one hop away from the meta-model $\theta$, making it easy to compute meta-gradients. Concretely, the meta-gradient descent for the meta model $\theta$ becomes $\theta \leftarrow \theta - \beta \mathbb{E}_{\mathcal{T} \sim P_{\mathcal{T}}, \mathcal{D}_{val} \sim \mathcal{T}} \nabla_\theta \mathcal{L}^{\mathcal{T}}_{\mathcal{D}_{val}}(\hat{\gamma}\theta + (1 - \hat{\gamma})\hat{\phi})$, which is apparently more manageable than the gradients in eq. (3) when $k > 1$.

**Algorithm 1** presents our "lazy" approach in detail. In the outer while-loop, we sample a batch of tasks $\{\mathcal{T}_i\}$ (Line 3, or L3) and use them to make a gradient update to the meta-model $\theta$ (L13). All task-specific models $\{\phi_{i,0}\}$ are initialized to the current meta-model $\theta$ (L6). For each task $\mathcal{T}_i$, the student first runs gradient descent with respect to the task-specific model $\phi_i$ up to $k$ steps (L8), and the teacher then takes a "leap" from the initial meta-model $\theta$ according to the checkpoints visited by the student (L10–11).

**Remarks.** Our "lazy" teacher is motivated by the lookahead optimizer (Zhang et al., 2019). They have some key differences as follows due to the meta-learning setup. We initialize multiple task-specific models by the meta-model. Moreover, we dynamically choose the "leap" rate $\gamma$ by a validation set. Finally, the validation data allows us to take advantage of not one checkpoint, but a region around the checkpoints visited by the student.

---

**Algorithm 1** "Lazy" Meta-Learning

**Require:** A distribution over tasks $P_{\mathcal{T}}$
**Require:** Learning rates $\eta, \beta$
**Ensure:** The meta model $\theta$
1: Randomly initialize the meta-model $\theta$
2: **while** not done **do**
3:      Sample a batch of tasks $\{\mathcal{T}_i \sim P_{\mathcal{T}}\}$
4:      **for all** $\{\mathcal{T}_i\}$ **do**
5:          Sample data $\mathcal{D}_{tr}$ and $\mathcal{D}_{val}$ for $\mathcal{T}_i$
6:          $\phi_{i,0} \leftarrow \theta$
7:          **for** $j = 1, 2, \cdots, k$ **do**      //student
8:              $\phi_{i,j} \leftarrow \phi_{i,j-1} - \eta \nabla_\phi \mathcal{L}^{\mathcal{T}_i}_{\mathcal{D}_{tr}}(\phi_{i,j-1})$
9:          **end for**
10:        Grid-search $\mathcal{M}_{\text{LAZY}}(\theta, \mathcal{D}_{tr})$ such that $\mathcal{L}^{\mathcal{T}_i}_{\mathcal{D}_{val}}$ is minimized at $\hat{\gamma}_i\theta + (1 - \hat{\gamma}_i)\hat{\phi}_i$ //teacher
11:          $\phi_i(\theta) \leftarrow \hat{\gamma}_i\theta + (1 - \hat{\gamma}_i)\hat{\phi}_i$    //teacher
12:      **end for**
13:      $\theta \leftarrow \theta - \beta \nabla_\theta \sum_i \mathcal{L}^{\mathcal{T}_i}_{\mathcal{D}_{val}}(\phi_i(\theta))$
14: **end while**

---

The teacher's role is similar to the skip connection in ResNet (He et al., 2016). They bridge two otherwise distant points such that gradients can effectively propagate between them.

We share the same goal, to make MAML less "greedy", as the recently proposed implicit gradients (iMAML) (Rajeswaran et al., 2019). iMAML changes the lower-level problem in eq. (1) to an $\ell_2$-regularized problem, which lends an analytical expression for the meta-gradient. But it is expensive to compute and has to be approximated by a conjugate gradient algorithm. The $\ell_2$ regularization also falls short in capturing structural relations between a meta-model and task-specific models.

## 3.2   Few-Shot Learning, Long-Tailed Classification, and Meta-Attack

Since the "lazy" teacher does not change the innermost loop of gradient-based meta-learning — it instead "leaps" over the chain of updates to the task-specific model $\phi$, we can apply it to different algorithms. We evaluate it on few-shot learning, long-tailed classification, and meta-attack, in which meta-learning based methods have led to state-of-the-art results.

**Few-shot learning** in this paper concerns an $N$-way-$K$-shot classification problem. To customize Algorithm 1 for this problem, we randomly select $N$ classes for each task $\mathcal{T}_i$ and then draw from each class $K + 1$ examples with labels, $K$ of which are assigned to the training set $\mathcal{D}_{tr}$ and one is to the validation set $\mathcal{D}_{val}$. Besides, we choose the hyper-parameter $\gamma_i$ by using the task-specific model's classification accuracy on the validation set, instead of the loss in L10, Algorithm 1.

There is an interesting "trap" in few-shot learning, identified as over-fitting by memorization (Yin et al., 2019). The tasks $\{\mathcal{T}_i\}$ drawn from a distribution $P_{\mathcal{T}}$ are supposed to be i.i.d., but they could be correlated in the following scenario. Suppose there exists a global order of all classes. If we maintain this order among the $N$ classes in each task, the meta-model could over-fit the tasks seen during meta-training by memorizing the functions that solve these tasks, and it would fail to generalize to new tasks. Hence, it is important to randomly shuffle the $N$ classes every time we sample them for a

---

**Algorithm 2** "Lazy" Two-Component Weighting for Long-Tailed Recognition

---

**Require:** A training set $\mathcal{D}_{tr}$ whose class frequency is long-tailed, a balanced validation set $\mathcal{D}_{val}$
**Require:** Class-wise weights $\{w_y\}$ estimated by using Cui et al. (2019)
**Require:** Learning rates $\eta, \tau$, pre-training steps $t_1$, fine-tuning steps $t_2$
 1: Train a recognition network, parameterized by $\theta$, for $t_1$ steps by a standard cross-entropy loss
 2: **for** $t = t_1 + 1, \cdots, t_1 + t_2$ **do**
 3:     Sample a mini-batch $B$ from the training set $\mathcal{D}_{tr}$
 4:     Set $\epsilon_i \leftarrow 0, \forall i \in B$, and denote by $\epsilon := \{\epsilon_i, i \in B\}$
 5:     Compute $\mathcal{L}_B(\theta, \epsilon) := \frac{1}{|B|} \sum_{i \in B} (w_{y_i} + \epsilon_i) \mathcal{L}_i(\theta)$   //$\mathcal{L}_i$ is a cross-entropy over the $i$-th input
 6:     Update $\tilde{\theta}(\epsilon) \leftarrow \theta - \eta \nabla_\theta \mathcal{L}_B(\theta, \epsilon)$           // The "lazy" teacher, which depends on $\epsilon$
 7:     Initialize a student model by setting $\phi_0 \leftarrow \tilde{\theta}(\epsilon)$
 8:     **for** $j = 1, 2, ..., k$ **do**
 9:         Update the student model by gradient descent $\phi_j \leftarrow \phi_{j-1} - \eta \nabla_\phi \mathcal{L}_B(\phi_{j-1}, \epsilon)$
10:     **end for**
11:     Grid search for $\gamma$ s.t. the teacher's "leap", $\gamma \tilde{\theta}(\epsilon) + (1 - \gamma)\phi_k$, yields high accuracy on $\mathcal{D}_{val}$
12:     Update $\epsilon \leftarrow \epsilon - \tau \nabla_\epsilon \mathcal{L}_{\mathcal{D}_{val}}(\gamma \tilde{\theta}(\epsilon) + (1 - \gamma)\phi_k)$
13:     Compute $\mathcal{L}_B(\theta, \epsilon)$ (cf. Line 5) and update $\theta \leftarrow \theta - \eta \nabla_\theta \mathcal{L}_B(\theta, \epsilon)$
14: **end for**

---

task (e.g., "dogs" and "cats" are respectively labeled as 0 and 1 in a two-way classification task, and yet they are shuffled to 1 and 0 in another two-way task).

We will empirically show that our approach is less prone to over-fitting than MAML even without class shuffling. A possible reason is that we use longer chains of updates ($\phi_0, \cdots, \phi_k, k > 10$) to learn the functions that solve the individual tasks, making them harder to memorize.

**Long-tailed classification** emerges as an inevitable challenge as object recognition makes progress toward large-scale, fine-grained classes (Van Horn et al., 2018; Weyand et al., 2020), which often exhibit a long-tailed distribution. To uplift infrequent classes, Jamal et al. (2020) propose to weigh each training example by two components, a fixed component $w_y$ to balance different classes (Cui et al., 2019) and a trainable component $\epsilon_i$. We improve their learning method by a "lazy" teacher, as described in Algorithm 2. It alternatively optimizes the per-example weight $\epsilon_i$ (using a balanced validation set) and a recognition network $\theta$ (using the long-tailed training set), in the same spirit as meta learning (cf. Algorithm 1 vs. L5-12 in Algorithm 2). We insert a "lazy" teacher model to L6, let it take a "leap" in L12, and then backpropagate the gradient with respect to the per-example weight $\epsilon_i$ through the "leap".

**Meta-attack** (Du et al., 2019) is a query-efficient blackbox attack algorithm on deep neural networks. Recent work has shown that one can manipulate an image recognition network's predictions by adding very small perturbations to benign inputs. However, if the network's architecture and weights are unknown (blackbox), it takes a large number of queries into the network to find a valid adversarial example. To improve the query efficiency, Du et al. (2019) propose to learn a meta-model from many whitebox neural networks and then generalize it to blackbox attacks. They train this meta-model by using the same meta-learning framework as Algorithm 1. Therefore, it is straightforward to improve their inner loop by our "lazy" teacher; we postpone the detailed algorithm to Appendix B.

## 4 EXPERIMENTS

We evaluate the "lazy", long-horizon meta-learning approach by plugging it into different algorithms with applications to few-shot learning, long-tailed recognition, and meta-attack.

### 4.1 FEW-SHOT LEARNING

We experiment with four datasets for few-shot learning: Omniglot (Lake et al., 2011), MiniImageNet (Vinyals et al., 2016b), TieredImageNet (Ren et al., 2018a), and CIFAR-FS (Bertinetto et al., 2018). The experiment protocols and implementation details largely follow MAML (Finn et al., 2017) and Reptile (Nichol et al., 2018). Please refer to Appendices A.1 and A.2 for more details.

Our approach permits long-horizon inner updates and involves a convex hull of the last few checkpoints. In Table 1, we first experiment with the last $b=3$ and $b=1$ checkpoints. We test them with two representative meta-learning algorithms: MAML (cf. Algorithm 1) and Reptile (replacing Line 13 (L13) in Algorithm 1 with $\theta \leftarrow \theta - \beta \sum_i (\theta - \phi_i(\theta))$). The intervals are 0.05 in the grid search (L10), and the search range for the learning rate $\gamma$ is between 0.75 and 0.95.

Table 1: Our approach applied to MAML and Reptile for five-way few-shot classification on MiniImageNet (Accuracy $\pm 95\%$ confidence interval over 2000 runs)

| Method | MiniImageNet | |
|---|---|---|
| | 1-shot | 5-shot |
| MAML (Finn et al., 2017) | $48.70 \pm 1.84$ | $63.11 \pm 0.92$ |
| "Lazy" MAML ($b = 1$) | $48.26 \pm 1.78$ | $64.13 \pm 1.90$ |
| "Lazy" MAML ($b = 3$) | $48.17 \pm 1.84$ | $63.73 \pm 1.10$ |
| Reptile (Nichol et al., 2018) | $49.97 \pm 0.32$ | $65.99 \pm 0.58$ |
| "Lazy" Reptile ($b = 1$) | $51.50 \pm 1.00$ | $67.22 \pm 0.97$ |
| "Lazy" Reptile ($b = 3$) | $52.67 \pm 1.01$ | $68.77 \pm 0.98$ |

Table 1 shows that there is no significant difference between $b = 3$ and $b = 1$, so we shall employ $b = 1$ for the remaining experiments. Moreover, the "lazy" variation improves the vanilla Reptile, but not MAML, probably because the five-way one/five-shot learning is too simple for MAML to take advantage of the long-horizon inner updates. We next study many-way few-shot learning tasks, which are arguably more complex.

### 4.1.1 MAML VS. "LAZY" MAML FOR MANY-WAY FEW-SHOT LEARNING

We switch to the TieredImageNet dataset since there are only 20 classes in MiniImageNet's meta-test set. The left panel of Figure 2 shows the results of MAML and "Lazy" MAML for $N$-way-five-shot learning, where $N$ varies in $\{5, 20, 30, 50\}$, and the student runs for $k = 10, 15, 20, 20$ inner steps, respectively. The "lazy" variation is on par with MAML for the five-way classification, and it significantly outperforms MAML for 20-way, 30-way, and 50-way five-shot classifications. This trend indicates that the many-way few-shot learning problems desire more inner updates to the task-specific models, amplifying the benefit of the "lazy" teacher.

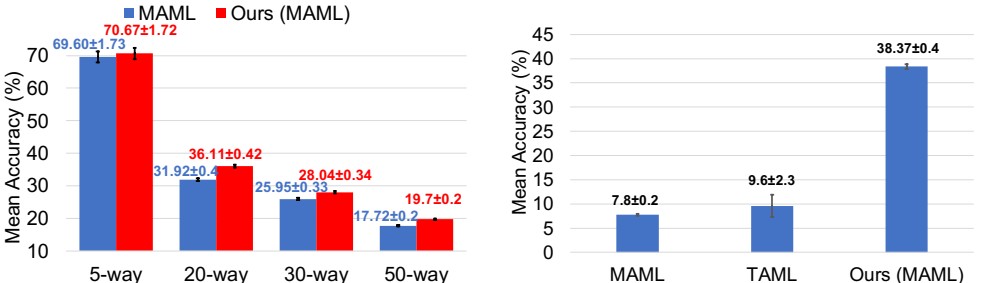

Figure 2: Left: Mean Accuracy (%) for $N$-way-five-shot classification on TieredImageNet. Right: Mean Accuracy (%) for 20-way-one-shot non-i.i.d. (Yin et al., 2019) classification tasks on Omniglot.

### 4.1.2 "LAZY" MAML IS LESS PRONE TO OVER-FITTING BY MEMORIZATION THAN MAML

The right panel of Figure 2 shows some 20-way-one-shot classification results on Omniglot when we learn from non-i.id. tasks, i.e., by maintaining a global order of all training classes. This global order creates a shortcut for meta-learning methods; they may memorize the order from the meta-training tasks and fail to generalize to meta-test tasks (Yin et al., 2019). We can see that the "lazy" teacher boosts MAML by a large margin and outperforms TAML (Jamal & Qi, 2019), indicating that it is less prone to over-fitting by memorization. A plausible reason is that the $k = 15$ steps taken by the exploratory student make it harder to memorize than the one-step update in MAML or TAML.

**In Appendix A**, we present more results of the few-shot learning. Section A.5 investigates the proposed "lazy" approach with Reptile-style update for $N$-way-five-shot learning on TieredImageNet. Section A.4 further compares MAML and "lazy" MAML by their computation memory costs. Section A.6 contrasts "lazy" MAML and "lazy" Reptile to prior arts on all the four datasets.

## 4.2 LONG-TAILED CLASSIFICATION

Following the experiment setup in (Cui et al., 2019) and (Jamal et al., 2020), we use the CIFAR-LT-100 dataset (Cui et al., 2019) to compare our Algorithm 2 with several long-tailed recognition methods. Cui et al. (2019) created multiple long-tailed datasets by removing training examples from CIFAR-100 (Krizhevsky & Hinton, 2009) according to different power law distributions. In each version, we compute an imbalance factor as the ratio between the sizes of the head class and the tail class. We run $k = 5$ steps in the innermost loop of Algorithm 2.

Table 2: Test top-1 errors (%) of ResNet-32 on CIFAR-LT-100 under different imbalance settings.

| Method ↓                                                  Imbalance factor → | 200 | 100 | 50 | 20 |
|---|---|---|---|---|
| Standard cross-entropy training | 65.16 | 61.68 | 56.15 | 48.86 |
| Class-balanced cross-entropy training (Cui et al., 2019) | 64.30 | 61.44 | 55.45 | 48.47 |
| Class-balanced fine-tuning (Cui et al., 2018) | 61.78 | 58.17 | 53.60 | 47.89 |
| Learning to reweight (Ren et al., 2018b) | 67.00 | 61.10 | 56.83 | 49.25 |
| Meta-weight (Shu et al., 2019) | 63.38 | 58.39 | 54.34 | 46.96 |
| Two-component weighting (Jamal et al., 2020) | 60.69 | 56.65 | 51.47 | 44.38 |
| **Lazy two-component weighting (ours)** | **58.67** | **53.46** | **48.24** | **43.68** |

Table 2 shows the test errors (%) under different imbalance factors. We can see that our teacher-student scheme boosts the original two-component weighting approach (Jamal et al., 2020) under all the imbalance factors. The results are especially interesting in that Algorithm 2 is not exactly a meta-learning method, though it shares the same framework as the gradient-based meta-learning due to the two nested optimization loops. Besides, compared with the other competing methods, our results establish a new state of the arts for the long-tailed object recognition.

## 4.3 META-ATTACK

We evaluate the "lazy" meta-attack on MNIST (LeCun et al., 1998) and CIFAR-10 (Krizhevsky & Hinton, 2009). We follow (Du et al., 2019) for the experiment setup and all training details, including the network architectures used to generate gradients for input images, the attack models, meta-attack models, and evaluation metrics for both the datasets, to name a few. The learning rates in the inner and outer loops are both 0.01. We let the student run $k = 8$ and $k = 10$ steps in the innermost loop for MNIST and CIFAR-10, respectively.

Table 3: Untargeted adversarial attack results on MNIST and CIFAR10. We achieve comparable success rates and average $\ell_2$ distortions with other methods by using a smaller number of queries.

| Dataset / Target model | Method | Success Rate | Avg. $\ell_2$ | Avg. Queries |
|---|---|---|---|---|
| MNIST / Net4 | Zoo (Chen et al., 2017) | 1.00 | 1.61 | 21,760 |
| | Decision boundary (* et al., 2018) | 1.00 | 1.85 | 13,630 |
| | Opt-attack (Cheng et al., 2019) | 1.00 | 1.85 | 12,925 |
| | AutoZoom (Tu et al., 2018), | 1.00 | 1.86 | 2,412 |
| | Bandits (Ilyas et al., 2019) | 0.73 | 1.99 | 3,771 |
| | Meta-attack (Du et al., 2019) | 1.00 | 1.77 | 749 |
| | **Lazy meta-attack (ours)** | 1.00 | 1.65 | **566** |
| CIFAR10 / Resnet18 | Zoo (Chen et al., 2017) | 1.00 | 0.30 | 8,192 |
| | Decision boundary (* et al., 2018) | 1.00 | 0.30 | 17,010 |
| | Opt-attack (Cheng et al., 2019) | 1.00 | 0.33 | 20,407 |
| | AutoZoom (Tu et al., 2018) | 1.00 | 0.28 | 3,112 |
| | Bandits (Ilyas et al., 2019) | 0.91 | 0.33 | 4,491 |
| | FW-black (Chen et al., 2018) | 1.00 | 0.43 | 5,021 |
| | Meta-attack (Du et al., 2019) | 0.94 | 0.34 | 1,583 |
| | **Lazy meta-attack (ours)** | 0.98 | 0.45 | **1,061** |

Table 3 shows the results of untargeted attack, namely, the attack is considered successful once it alters the recognition network's prediction to any incorrect class. Appendix B includes the results of targeted attack. In addition to the original meta-attack (Du et al., 2019), Table 3 also presents

several existing blackbox attack methods for comparison. We can see that meta-attack and our "lazy" meta-attack yield about the same success rates as the other blackbox attacks. The second-to-the-right column is about the average $\ell_2$ distortion an attacker makes to an input, the lower the better. The rightmost column is about the number of queries an attacker makes into the recognition network, the lower the better. The "lazy" meta-attack is able to achieve comparable success rates and $\ell_2$ distortion rates with the other methods yet by using a smaller number of queries. Both meta-attack and its "lazy" version significantly outperform the other methods in terms of the query efficiency, indicating the generalization capability of the meta-attack model from known whitebox neural networks to unknown blackbox networks.

## 5 RELATED WORK

Meta-learning has been a long-standing sub-field in machine learning (Schmidhuber, 1987; Thrun & Pratt, 1998; Naik & Mammone, 1992). Early approaches update a model's parameters by training a meta-learner (Bengio et al., 1995; Bengio et al., 1991; Schmidhuber, 1992). This has been well studied in optimizing neural networks, and one such family of meta-learning learns an optimizer (Ravi & Larochelle, 2016; Li & Malik, 2016; Andrychowicz et al., 2016). A specialized neural network takes gradients as input and outputs an update rule for the learner. In addition to the update rule, Ravi & Larochelle (2016) also learn the weight initialization for few-shot learning. Finally, there are several approaches (Metz et al., 2019; Wichrowska et al., 2017) for training generic optimizers that can be applied broadly to different neural networks and datasets.

Under the context of few-shot learning, another family of meta-learning involves metric-learning based methods (Vinyals et al., 2016a; Snell et al., 2017; Mishra et al., 2017; Koch et al., 2015; Oreshkin et al., 2018), which learn a metric space to benefit different few-shot learning tasks. The goal is to find the similarity between two samples regardless of their classes using some distance metric so that the similarity function can be used to classify the unseen classes at the test stage. Some recent studies along this line include Matching Networks (Vinyals et al., 2016a), which employs the cosine similarity, Prototypical Networks (Snell et al., 2017), which uses the Euclidean distance to compute the similarity, Relation Network (Sung et al., 2017), which uses a relation module as the similarity function, ridge regression (Bertinetto et al., 2018), and graph neural networks (Satorras & Estrach, 2018).

More recently, gradient-based meta-learning gains its momentum, and a variety of methods have been proposed in this vein. The most notable one among them might be MAML (Finn et al., 2017), where the goal is to learn the network weight initialization so that it can adapt to unseen tasks rapidly. There have been extensions to improve MAML. Meta-SGD (Li et al., 2017) learns the learning rates along with the weight initialization. Regularization techniques (Yin et al., 2020; Jamal & Qi, 2019) are introduced to MAML to mitigate over-fitting. (Park & Oliva, 2019) preconditions on the gradients in the inner loop by learning a curvature. Despite MAML's popularity, it is still computationally expensive and consumes large memory due to the computation of high-order derivatives. The authors show that the first-order approximation, which neglects the gradients of the inner loop during meta-optimization, performs about the same as the original MAML. Another first-order meta-learning method is Reptile (Nichol et al., 2018), which decouples the inner and outer optimization steps. iMAML (Rajeswaran et al., 2019) provides an approximate solution for meta-gradients by using an algorithm based on conjugate gradients, and its low-level optimization is similar to Meta-MinibatchProx (Zhou et al., 2019). The idea is to add an $\ell_2$ regularizer in the inner loop, allowing the updated parameters close to the initial parameters.

## 6 CONCLUSION

We propose a teacher-student scheme for the gradient-based meta-learning algorithms to allow them run more steps of inner updates to task-specific models while being immune to the risk of vanishing or exploding gradients. The student explores the tasks-specific model's feasible space up to many steps, and the "lazy" teacher takes a one-step "leap" towards the region explored by the student. As a result, the teacher defines a lightweight computation graph and yet it takes advantage of the adequately explored checkpoints by the student. This approach is generic; we apply it to different problems, include few-shot learning, long-tail recognition, and meta-attack and various meta-learning methods. Experiments verify the benefit of long-horizon inner updates in gradient-based meta-learning.

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

## APPENDICES

## A FEW-SHOT LEARNING

### A.1 DATASETS

We experiment with four datasets for few-shot learning: Omniglot (Lake et al., 2011), MiniImageNet (Vinyals et al., 2016b), TieredImageNet (Ren et al., 2018a), and CIFAR-FS (Bertinetto et al., 2018). The Omniglot dataset consists of handwritten characters from 50 different alphabets and 1623 characters. There are 20 handwritten examples of each character. MiniImageNet contains 100 classes form ImageNet (Deng et al., 2009), which are split to 64, 16, and 20 classes for meta-training,

meta-validation, and meta-test, respectively. TieredImagNet has 608 classes from ImageNet, which are grouped into 34 higher-level categories following the ImageNet taxonomy. They are split into 20 meta-training categories, 6 meta-validation categories, and 8 meta-test categories. Due to this partition scheme, the meta-test classes are less similar to the meta-training classes in TieredImageNet than in other datasets. CIFAR-FS re-purposes CIFAR-100 (Krizhevsky & Hinton, 2009), splitting its 100 classes into 64, 16, and 20 classes for meta-training, meta-validation, and meta-test, respectively.

### A.2    EXPERIMENT PROTOCOLS AND HYPER-PARAMETERS

Our experiment protocols and implementation details largely follow MAML (Finn et al., 2017) and Reptile (Nichol et al., 2018). In particular, we use a convolutional neural network that comprises four modules in all the experiments. Each module has 3x3 convolutions, a batch-normalization layer, 2x2 max-pooling, and the ReLU activation, and every convolutional layer contains 64 filters for the experiments on Omniglot and 32 filters for other datasets. For fair comparison, we also re-implement some of the existing methods using this network architecture. We report more details for the "Lazy" Reptile in Table 4. For "Lazy" MAML, we set $k = 10$ for 1-shot and 5-shot tasks, respectively, on both MiniImageNet and TieredImageNet.

Table 4: Hyper-parameter details for few-shot learning in ours (Reptile). The "Eval inner batch" row shows the numbers for both 1-shot and 5-shot settings.

| Hyper-parameter | Omniglot | CIFAR-FS | Mini-ImageNet | TieredImageNet |
|---|---|---|---|---|
| Inner learning rate ($\eta$) | 0.001 | 0.001 | 0.001 | 0.001 |
| Inner iterations ($k$) | 5 | 8 | 8 | 8 |
| Inner batch size | 10 | 10 | 10 | 10 |
| Training shots | 10 | 15 | 15 | 15 |
| Outer step-size ($\beta$) | 1.0 | 1.0 | 1.0 | 1.0 |
| Total outer-iterations | 100k | 120k | 120k | 130k |
| Meta batch size | 20 | 20 | 20 | 20 |
| Eval. inner iterations | 50 | 50 | 50 | 50 |
| Eval. inner batch | 5/15 | 5/15 | 5/15 | 5/15 |

### A.3    MANY-WAY CLASSIFICATION

We have presented many-way results on TieredImageNet for "Lazy" MAML in section 4.1.1. Here, we describe the implementation details for these experiments. We set the inner learning rate ($\eta$) to 0.005 and the outer learning rate ($\beta$) to 0.001 for all the settings. We let the student run $k = 15$, 15 and 18 steps for 20-way, 30-way and 50-way, respectively during meta-training.

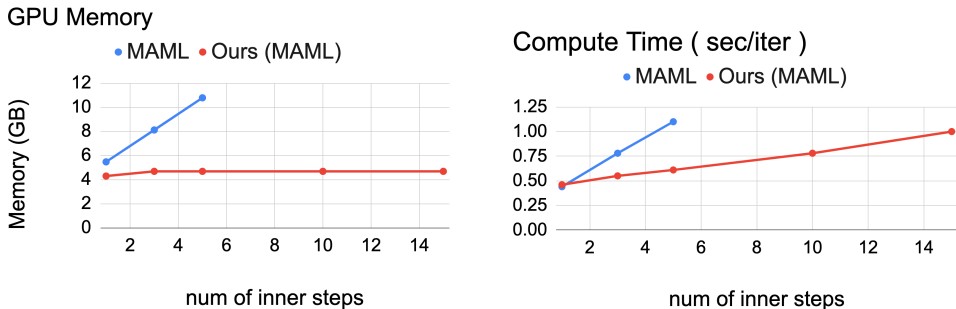

Figure 3:   Left: Memory trade-offs with 4 layer CNN on 20-way-5-shot MiniImageNet task. b). Computation time (sec per meta-iteration) w.r.t the number of inner gradient steps on 20-way-5-shot MiniImageNet task.

### A.4 COMPUTATIONAL ANALYSIS

In our evaluation, we also want to answer the following question empirically. How does the memory and computation requirements of "Lazy" MAML compare with MAML? Figure 3 shows the memory and compute trade-off for "Lazy" MAML and MAML on 5-shot 20-way MiniImageNet. "Lazy" MAML decouples the dependency of inner and outer loop by teacher-student scheme which allows it to define a very lightweight computation graph. The left panel of the figure shows that the memory of the "Lazy" MAML doesn't exceed beyond 5 GB for many inner gradient steps while on the other hand, MAML reaches the capacity of 12 GB after 5 inner steps. The right panel shows the computation time per iteration with respect to multiple gradient inner steps. The time taken by "Lazy" MAML doesn't increase exponentially as compared to MAML which takes more compute time and reaches the maximum capacity of memory after 5 inner steps.

### A.5 ADDITIONAL RESULTS ON MANY-WAY LEARNING

The left panel of Figure 4 compares the results of Reptile and "Lazy" Reptile for $N$-way-five-shot learning on TieredImageNet where $N$ varies in $\{5, 20, 30\}$. Our approach outperforms Reptile. We emphasize that not all meta-learning algorithms can be approximated by a first-order version; for example, it is not immediately clear how to do it for Algorithm 2, the two-component weighting method for long-tailed classification, in the main text.

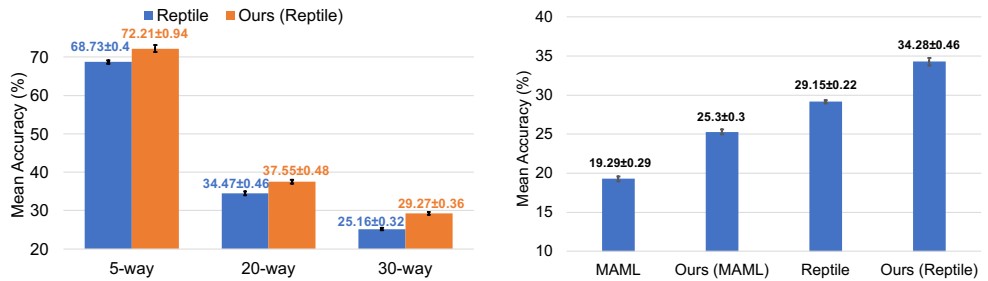

Figure 4: Left: Mean Accuracy (%) for $N$-way-five-shot classification on TieredImageNet. b). Mean Accuracy (%) for 20-way-5-shot classification on MiniImageNet.

The right panel of Figure 4 shows some 20-way-5-shot results on MiniImageNet. We can see that our lazy strategy boosts both MAML and Reptile by a significant margin, which is similar to what we see in Figure 2 of the main paper. It again indicates that more training data needs more steps of exploration for a task-specific model and hence magnifies the benefit of our teacher-student scheme introduced to both MAML and Reptile.

### A.6 COMPARISON RESULTS

We compare "our" approach with state-of-the-art meta-learning methods for five-way few-shot learning problems on four datasets. The results are shown in Tables 5 and 6. For our own approach, we study both the MAML-style update to the meta-model (ours (MAML), L13 in Algorithm 1) and the Reptile-style (Nichol et al., 2018) update (ours (Reptile), L14 in Algorithm 1) for MiniImageNet and TieredImageNet. On Omniglot and CIFAR-FS, we only report ours (Reptile) due to its low computation cost. Batch normalization with test data yields about 2% improvement over the normalization with the training data only, and we report the results of both scenarios.

It can be seen that our results are better than or comparable with those of the competing methods. In general, the improvements by our teacher-student scheme are more significant on 5-shot settings than on 1-shot settings, verifying the trend in Section 4.1.1 that more training data can better leverage the exploratory student in our method. Besides, ours (Reptile) outperforms ours (MAML) probably for two reasons. One is that ours (Reptile) uses more than $k$ shots of training examples per class for a $k$-shot learning problem during meta-training, following the experiment setup of Reptile (Nichol et al., 2018). The other is that the second-order gradients in ours (MAML) make the training procedure less stable than Reptile. We hypothesize that a many-shot setting would be less sensitive to both

Table 5: Five-way few-shot classification accuracies (%) on MiniImageNet and TieredImageNet. The $\pm$ shows 95% confidence intervals computed over 2000 tasks.

| Method | BN w/ Test | Mini-ImageNet | | TieredImageNet | |
|---|---|---|---|---|---|
| | | 1-shot | 5-shot | 1-shot | 5-shot |
| MAML (Finn et al., 2017) | ✗ | $46.21 \pm 1.76$ | $61.12 \pm 1.01$ | $49.60 \pm 1.83$ | $66.58 \pm 1.78$ |
| MAML (Finn et al., 2017) | ✓ | $48.70 \pm 1.84$ | $63.11 \pm 0.92$ | $51.67 \pm 1.81$ | $69.60 \pm 1.73$ |
| Meta-Curvature (Park & Oliva, 2019) | ✓ | $48.83 \pm 1.80$ | $62.63 \pm 0.93$ | $50.30 \pm 1.99$ | $66.14 \pm 0.95$ |
| iMAML (Rajeswaran et al., 2019) | ✓ | $49.30 \pm 1.88$ | - | - | - |
| **Ours (MAML)** | ✓ | $48.26 \pm 1.78$ | $64.13 \pm 1.90$ | $51.03 \pm 1.70$ | $70.67 \pm 1.72$ |
| FOMAML (Finn et al., 2017) | ✗ | $45.53 \pm 1.58$ | $61.02 \pm 1.12$ | $48.01 \pm 1.74$ | $64.07 \pm 1.72$ |
| Reptile (Nichol et al., 2018) | ✗ | $47.07 \pm 0.26$ | $62.74 \pm 0.37$ | $49.12 \pm 0.43$ | $65.99 \pm 0.42$ |
| Meta-MinibatchProx (Zhou et al., 2019) | ✗ | $47.81 \pm 1.00$ | $63.18 \pm 1.00$ | $49.97 \pm 0.93$ | $66.60 \pm 0.91$ |
| **Ours (Reptile)** | ✗ | $48.14 \pm 0.94$ | $64.64 \pm 0.92$ | $51.15 \pm 0.95$ | $68.84 \pm 0.90$ |
| FOMAML (Finn et al., 2017) | ✓ | $48.07 \pm 1.75$ | $63.15 \pm 0.91$ | $50.12 \pm 1.82$ | $67.43 \pm 1.80$ |
| Reptile (Nichol et al., 2018) | ✓ | $49.97 \pm 0.32$ | $65.99 \pm 0.58$ | $51.34 \pm 0.4$ | $68.73 \pm 0.40$ |
| Meta-MinibatchProx (Zhou et al., 2019) | ✓ | $50.08 \pm 1.00$ | $66.28 \pm 0.98$ | $53.71 \pm 1.04$ | $69.78 \pm 0.95$ |
| **Ours (Reptile)** | ✓ | $\mathbf{51.50 \pm 1.00}$ | $\mathbf{67.22 \pm 0.97}$ | $\mathbf{54.41 \pm 1.00}$ | $\mathbf{72.21 \pm 0.94}$ |

Table 6: Five-way few-shot classification accuracies (%) on Omniglot and CIFAR-FS. The $\pm$ shows 95% confidence intervals computed over 1000 tasks.

| Method | BN w/ Test | Omniglot | | CIFAR-FS | |
|---|---|---|---|---|---|
| | | 1-shot | 5-shot | 1-shot | 5-shot |
| MAML (Finn et al., 2017) | ✓ | $98.70 \pm 0.40$ | $\mathbf{99.90 \pm 0.10}$ | $56.50 \pm 1.90$ | $70.50 \pm 0.90$ |
| iMAML (Rajeswaran et al., 2019) | ✓ | $\mathbf{99.16 \pm 0.35}$ | $99.67 \pm 0.12$ | - | - |
| Reptile (Nichol et al., 2018) | ✗ | $95.39 \pm 0.09$ | $98.90 \pm 0.10$ | $53.12 \pm 1.34$ | $69.40 \pm 1.30$ |
| **Ours (Reptile)** | ✗ | $95.44 \pm 0.57$ | $98.92 \pm 0.29$ | $54.64 \pm 1.30$ | $70.56 \pm 1.20$ |
| FOMAML (Finn et al., 2017) | ✓ | $98.30 \pm 0.50$ | $99.20 \pm 0.20$ | - | - |
| Reptile (Nichol et al., 2018) | ✓ | $97.68 \pm 0.04$ | $99.48 \pm 0.06$ | $57.50 \pm 0.45$ | $71.88 \pm 0.42$ |
| **Ours (Reptile)** | ✓ | $98.20 \pm 0.38$ | $99.70 \pm 0.16$ | $\mathbf{59.36 \pm 1.44}$ | $\mathbf{74.90 \pm 1.28}$ |

factors. Indeed, we verified this hypothesis by another five-way-50-shot learning experiment with ours (Reptile), which yields $76.17 \pm 0.32\%$ on MiniImageNet and is lower than $78.54 \pm 0.70$ by ours (MAML).

## B   META-ATTACK

Here, we formally present the algorithm of our lazy meta-learning approach to training the meta-attacker in Algorithm 3. As described in the main paper that the original meta-attack Du et al. (2019) uses Reptile to train the attacker, so it is straightforward to improve the approach by using a "lazy" teacher. The inputs to the meta-attacker are images, and the desired outputs are their gradients — during meta-training, the gradients are generated from different classification models. Instead of the cross-entropy loss, meta-attack adopts a mean-squared error (MSE) loss in the inner loop, i.e.,

$$\mathcal{L}_{\mathcal{D}_{tr}}^{\mathcal{T}_i} = ||\phi(\mathcal{X}_{ij}) - \mathcal{G}_{ij}||_2^2 \tag{6}$$

where the task $\mathcal{T}_i$ is to find adversarial examples for the inputs to the $i$-th pre-trained classification network, $\mathcal{X}_{ij}$ is an image sampled for the task, $\mathcal{G}_{ij}$ are the gradients of the classification network with respect to (w.r.t.) the image, and $\phi(\cdot)$ is a task-specific model whose output is to approximate the gradients $\mathcal{G}_{ij}$. This model is useful because, given a blackbox classification network, we can use the task-specific model to predict the gradients of this network w.r.t. an image, followed by gradient ascent towards an adversarial example (cf. Algorithm 4).

Algorithm 3 presents how to train this meta-attacker by applying our "lazy" teacher to Reptile, and we then follow Algorithm 4 for attacking blackbox networks Du et al. (2019).

**Algorithm 3** Training algorithm of meta-attack using "lazy" Reptile

---

**Require:** A distribution over tasks $P_{\mathcal{T}}$
**Require:** Input Images $\mathcal{X}$, gradients $\mathcal{G}_i$ generated from a classification network serving task $\mathcal{T}_i$
**Require:** Learning rates $\alpha, \beta$
**Ensure:** The meta attacker $\theta$
 1: Randomly initialize the meta-attacker $\theta$
 2: **while** not done **do**
 3:     Sample a batch of tasks $\{\mathcal{T}_i \sim P_{\mathcal{T}}\}$
 4:     **for all** $\{\mathcal{T}_i\}$ **do**
 5:         Sample data $\mathcal{D}_{tr}$ and $\mathcal{D}_{val}$ for $\mathcal{T}_i$                    // in the form of $\{\mathcal{X}_{ij}, \mathcal{G}_{ij}\}$
 6:         $\phi_{i,0} \leftarrow \theta$
 7:         **for** $j = 1, 2, \cdots, k$ **do**
 8:             $\phi_{i,j} \leftarrow \phi_{i,j-1} - \alpha \nabla_\phi \mathcal{L}_{\mathcal{D}_{tr}}^{\mathcal{T}_i}(\phi_{i,j-1})$
 9:         **end for**
10:         $\gamma_i \leftarrow \arg\min_\gamma \mathcal{L}_{\mathcal{D}_{val}}^{\mathcal{T}_i}(\gamma\theta + (1-\gamma)\phi_{i,k})$
11:         $\phi_i(\theta) \leftarrow \gamma_i\theta + (1 - \gamma_i)\phi_{i,k}$
12:     **end for**
13:     $\theta \leftarrow \theta - \beta \sum_i(\theta - \phi_i(\theta))$
14: **end while**

---

**Algorithm 4** Adversarial Meta-Attack

---

**Require:** Test image $x_o$ with label $t$, meta-attacker $f_\theta$, target model $\mathcal{M}_{tar}$, iteration interval $j$, selected top-$q$ coordinates
 1: **for** $t = 0, 1, 2, \cdots$ **do**
 2:     **if** $(t+1) \mod j = 0$ **then**
 3:         Perform zeroth-order gradient estimation on top-$q$ coordinates, denoted as $I_t$ and
 4:         obtain $g_t$.
 5:         Fine-tune meta-attacker $f_\theta$ with $(x_t, g_t)$ on $I_t$ by $\mathcal{L} = |[f_\theta(x_t)]_{I_t} - [g_t]_{I_t}|_2^2$.
 6:     **else**
 7:         Generate the gradient map $g_t$ directly from meta-attacker $f_\theta$ with $x_t$,
 8:         select coordinates $I_t$.
 9:     **end if**
10:     Update $[x']_{I_t} = [x_t]_{I_t} + \lambda[g_t]_{I_t}$.
11:     **if** $\mathcal{M}_{tar}(x') \neq t$ **then**
12:         $x_{adv} = x'$
13:         break
14:     **else**
15:         $x_{t+1} = x'$
16:     **end if**
17: **end for**
**Ensure:** adversarial example $x_{adv}$.

---

## B.1 Results under targeted attack

In Table 3 of the main paper, we report the results under untargeted attack. Here, we are presenting the results under targeted attack for both MNIST and CIFAR-10 in Table 7. Similar to untargeted attack, we achieve comparable results on success rate and average $\ell_2$ distortion using a smaller number of queries.

Table 7: Comparison of several methods under targeted attack on MNIST and CIFAR-10. Similar to the untargeted attack, we reduce the number of queries for meta attack.

| Dataset / Target model | Method | Success Rate | Avg. $L_2$ | Avg. Queries |
|---|---|---|---|---|
| MNIST / Net4 | Zoo Chen et al. (2017) | 1.00 | 2.63 | 23,552 |
| | Decision Boundary * et al. (2018) | 0.64 | 2.71 | 19,951 |
| | AutoZoom Tu et al. (2018) | 0.95 | 2.52 | 6,174 |
| | Opt-attack Cheng et al. (2019) | 1.00 | 2.33 | 99,661 |
| | Meta attack Du et al. (2019) | 1.00 | 2.66 | 1,299 |
| | **Lazy meta-attack (ours)** | 1.00 | 2.63 | **1,108** |
| CIFAR10 / Resnet18 | Zoo Chen et al. (2017) | 1.00 | 0.55 | 66,400 |
| | Decision Boundary * et al. (2018) | 0.58 | 0.53 | 16,250 |
| | AutoZoom Tu et al. (2018) | 1.00 | 0.51 | 9,082 |
| | Opt-attack Cheng et al. (2019) | 1.00 | 0.50 | 121,810 |
| | FW-black Chen et al. (2018) | 0.90 | 0.73 | 6,987 |
| | Meta attack Du et al. (2019) | 0.93 | 0.77 | 3,667 |
| | **Lazy meta-attack (ours)** | 0.92 | 0.69 | **3,092** |

