# OpenReview forum: "A Lazy Approach to Long-Horizon Gradient-Based Meta-Learning"
_ICLR.cc/2021/Conference — Reject_

### Official Review · AnonReviewer2 · 2020-10-26

**Rating:** 5
**Confidence:** 4

**Review:**

## Summary

The paper proposes a method to address the short horizon limitation of gradient based meta-learning algorithms. To avoid the multiple problems of back-propagating the gradient through inner loops with many steps, the authors propose a teacher-student solution that decouples the exploration (done by the student) from the actual computation of the gradient (done by the teacher). In particular, the inner loop is run using the student on the training set. The teacher then optimises the validation loss using grid search in a reduced search space defined by the convex hull of last points explored by the student and the common initialization. Finally, the metagradient is taken with respect to the initialization avoiding the backpropagation through the inner loop. In the experimental section, the authors apply the methods in three different use cases (few shot learning, long-tail classification and meta-attack)

## Comments

The paper is well motivated and the writing is clear.

From the technical perspective, the model proposal seems reasonable: focus on the gradient information toward the end of the trajectories defined by the task specific inner loops. One of the confusing aspect of the paper is that despite the general description of the method based on the convex hull of the last "b" updates of the inner loop, in the end only the last update is used (according to the remark at the end of section 4.1). If there is not advantage in performance by taking b>1 and assuming there is a significant cost in increasing b (the computational complexity of grid search grows exponentially with the dimensionality b), why to include this extension in the first place? Unless I am missing something it may make sense to start with the description of the simpler algorithm that uses only the last update.

I would also encourage the authors to elaborate on the similarities/differences between Reptile and their method in the same way they do it with MAML. In particular, when b=1 their method seems to be pretty close to Reptile (reptile uses \theta \leftarrow \theta + \epsilon(\widehat{\phi} -\theta) = (1 - \epsilon) \theta + \epsilon \widehat{\phi}), and so a deeper analysis of why the results are better would add value to the paper.

Regarding the experimentation, i would recommend the authors to move some of the comparisons with state-of-the-art methods from the appendix back to the manuscript. In particular, given that it is mentioned in Figure 1, I would include iMAML results in the manuscript rather than the appendix. Also, in the experimental setting, it is unclear why iMAML and FOMAML are incomplete in tables 5 and 6.

Finally, one of the relevant baselines that the authors missed is [1], which also addresses the short horizon bias by minimising the total length of the inner trajectories avoiding higher-order derivatives.

## Minors

* Figure 2: TAML is not defined
* Figure 1: The figure is quite confusing and it is not clear what the authors try to convey (e.g. should not be the task specific gradient be pointing in the same direction across the three plots?). Why is iMAML included but not Reptile, when it is actually Reptile and not iMAML the one baseline that appears in the main manuscript.
* At the end of page 4, the authors mention the importance of shuffling the N classes and then that their method performs better than MAML even without shuffling. Does that mean that all the experiments (for all methods) are done without shuffling? How do the results vary when shuffling is added?
* Equation 4: k-b+1...k should be k-b+j for j in {1,k}

## References
[1] Transferring Knowledge across Learning Processes

---

> ### Author Response · Authors · 2020-11-22
> **Response to R2 Questions**
>
> Thank you for the detailed review and constructive feedback! Please find our responses to the questions raised in the review as follows.
>
> >  **why to include this (convex hull) extension in the first place?**
>
> We planned to have the approach generic and applicable to few-shot learning, long-tailed classification, meta-attack, and others that may benefit from the gradient-based meta-learning. Hence, we presented the general form with a convex hull, preparing for future applications. Thanks for letting us know that it incurs some level of confusion; we will present $b=1$ first, followed by a discussion about $b>1$, in the revised paper.
>
> > **elaborate on the similarities/differences between Reptile and their method**
>
> We will include the following in the revised paper. Like Reptile, our approach allows the inner loop to make many steps of updates to task-specific models. Moreover, as the reviewer pointed out, we share the same update rule as Reptile by the end of the many-step exploration. However, we apply that rule to the task-specific models, while Reptile essentially uses it to update the meta-model. Unlike Reptile, we use meta-gradients to update the meta-model. This difference is subtle and vital, making it straightforward to apply our approach to the two-component weighting algorithm for long-tailed classification (and other meta-like algorithms) but unclear how to do it for Reptile. The difference also makes it possible for us to optimize the learning rate using task-specific models' validation data.
>
> > **Suggestion for reorganizing the experiments**
>
> We are thankful for the suggestion. We will reorganize the experiment section in the revised paper.
>
> The iMAML numbers are from existing papers since we did not find any publicly available source code, and we could not reproduce a reliable implementation ourselves. We will include FOMAML in Tables 5 and 6. New experiments with FOMAML on CIFAR-FS show that it achieves 55.6 ± 1.88 and 69.52 ± 0.91 for 1-shot-5-way and 5-shot-5-way learning, respectively, which are lower than ours (59.36 ± 1.44 and 74.90 ± 1.28).
>
> > **[1] also addresses the short horizon bias by minimizing the total length of the inner trajectories avoiding higher-order derivatives.**
>
> We are again thankful for the suggestion of the related work. We will cite and discuss this paper.
>
> We follow the authors’ code at https://github.com/amzn/metalearn-leap, and run it on MiniImageNet for 1-shot- 5-way learning. It gives rise to 42.53 ±1.69, which is lower than the original MAML (and ours). We do find that the method is memory efficient, but it requires more training time than the first order methods like FOMAML and, Reptile.
>
> > **Minors**
>
> * TAML is an entropy regularization method to reduce meta-overfitting. We will explain it in the revised paper.
> * We borrowed Figure 1 from the iMAML paper and added an illustration of our approach. It is a good idea to add Reptile to the figure, too. The task-specific gradients are supposed to point to different directions because MAML runs the inner loop for only one or a few steps, iMAML runs it almost till convergence, while ours executes it for many ($k$) steps.
> * No shuffling is used only in the memorization experiments (Section 4.1.2).
> * We will make equation (4) more clear.

---

### Official Review · AnonReviewer3 · 2020-10-29
**Simple approach to alleviate higher-order gradient computation**

**Rating:** 7
**Confidence:** 3

**Review:**

Computation of higher-order gradients is a major bottleneck in gradient-based meta-learning algorithms like MAML. This work proposes a simple approximation algorithm to alleviate the issue. They use a student network to explore the search space of task-specific models. Unlike MAML, the computation graph of the task-specific adaptation is not tracked and this allows the number of task-specific updates to be larger. Once the task-specific adaptation is done, they simply take a convex combination of the task-adapted parameters and the original parameters and evaluate the resulting model on the validation data of the task. The validation error is then backprogated for training. The proposed approach seems to be applicable to different meta-learning scenarios including few-shot learning, unbalanced classification and meta-attack. On the unbalanced classification and the meta-attack benchmarks, they showed marginal improvements over existing methods.

Pros:
- The paper is well written and clear
- They evaluated the algorithm on multiple different meta-learning problems
Strong results on the unbalanced classification and the meta-attack benchmarks

Cons:
- Although the approach is motivated by the look ahead optimization, it feels a bit unintuitive to take a simple convex combination of the adapted and the original parameters for the sake of skip connection
- I am not fully convinced that the method alleviates the short-horizon biases of the gradient-based meta-learning algorithm. The meta-update still seems to favor a solution that is closest to the original model due to the skip connection.
- Evaluation on multi-shot learning benchmarks can make the work strong

---

> ### Author Response · Authors · 2020-11-25
> **Response to R3 Questions**
>
> Thank you for the detailed review and constructive feedback! Please see our responses to each of the “Cons” bullet points as follows.
>
> * The intuition is to use lookahead to build a bridge between a task-specific model's initialization and its endpoints after a long-horizon exploration. Unlike the original lookahead, we search for the learning rates and other hyper-parameters by the task-specific validation set --- made possible thanks to the meta-learning setup. If we unroll the optimization iterations into a multi-layer neural network, the teacher's leap essentially becomes a skip connection between different layers.
>
> * The student explores the feasible space by a large number of steps, which are expensive or even infeasible in conventional gradient-based meta-learning (GBMT, e.g., MAML), which are bounded by memory and computation constraints. Since the skip connection is to the endpoints after a long-horizon exploration, it "favor(s) ... the original model" less than existing GBMT methods.
>
> * Thanks for the suggestion! We have run experiments for the multi-shot settings on MiniImageNet. We compare lazy MAML to the original MAML on 20-shot and 50-shot for 5-way classification. For 20-shot, the lazy MAML achieves 74.8 ± 0.76, while MAML yields 66.18 ± 0.7. For 50-shot, the lazy MAML leads to 78.54 ± 0.7, while MAML produces 67.13 ± 0.78. It shows the benefit of the exploratory student in our approach, where we let the student run for $k = 15, 20$ steps for the 20-shot and 50-shot settings, respectively. We'll include the experiments in the revised paper.

---

### Official Review · AnonReviewer4 · 2020-10-30
**Review of "Lazy" GBML**

**Rating:** 5
**Confidence:** 4

**Review:**

This paper studies the problem setting of gradient based meta learning (GBML) and presents a new algorithmic approach that is computationally tractable. The proposed approach follows a two time-scale algorithm, where the inner level takes multiple gradient steps of the task loss function, while the outer level takes smaller "leaps" following a weighted average of the inner level iterates. Experimental results are provided for few-shot classification and query efficient "meta attack" on deep networks which suggest that the proposed algorithms perform favorably when compared to baselines.

**Assessment**

The proposed algorithm and motivations are interesting. The experimental results are convincing. However, the writing lacks clarity (some of these are below). Overall, my recommendation is borderline. I am happy to reconsider based on author response.

**Questions and Comments**

(1) Having very long inner loops can lead to divergence in the bi-level optimization setup, since the inner level parameters are no longer influenced by the meta-parameters. For example, if sufficient number of steps are used, the inner level will converge to a stable point regardless of initialization. This is related to the vanishing gradient problem and is explained in greater detail in the iMAML paper. This suggests that we either need early stopping (like in MAML) or some other explicit regularization (like in iMAML). How does your problem formulation consider this problem and how do you propose to use long inner loop training without any explicit or implicit regularization?

(2) The paper writes $\phi_i(\theta) = \hat{\gamma}_i \theta + (1-\hat{\gamma}_i) \hat{\phi}_i$ where $\hat{\phi}_i$ is a convex combinations of iterations in SGD for task $i$. Is $\hat{\phi}_i$ considered to be a function of $\theta$, or do you pass stop-grad through the computation? This seems like a crucial detail that is missing, since if stop grad is not used, then one would still be differentiating through the optimization path. If stop grad is indeed used, why does the proposed algorithm solve the bi-level optimization?

(3) "The teacher’s role is similar to the skip connection in ResNet (He et al., 2016). They bridge two otherwise distant points such that gradients can effectively propagate between them." -- this seems like an unsubstantiated claim. In ResNets, gradients are passed through the skip connections. Do you also do this (see pt. 2 above)? Secondly,  ResNets perform skip connections from input to output, whereas GBML deals with optimization iterates. In my opinion, the correct analogy to ResNets would look something like Nesterov's momentum (where previous iterates are used to find the next iterate).

(4) Lazy MAML requires hyper-parameter optimization over the different $\alpha_i$ and $\gamma$ for each **meta-training iteration**. This presumably would add significant computational overhead compared to MAML. This contradicts with the compute results in Appendix A.3. The cost of back-propagation through a graph is typically no more than 4-5 times as forward prop (regardless of the graph). If 4-5 hyperparameter choices are tried out, then the number of forward prop in case of lazy MAML would exceed this compute factor for backprop. Can the authors clarify as to why lazy MAML is more compute efficient?

(5) Finally, I believe the authors may have a misunderstanding about MAML and related algorithms. Having a longer inner loop does not imply that higher order gradients are needed. As long as SGD is the inner loop optimizer, regardless of the size of the inner loop, only Hessian-vector products would be needed. See the iMAML paper for further discussion about these aspects. However, the memory footprint will increase linearly with the length of the inner loop. The authors of the submission point out the memory drawback correctly, but IMHO are a bit imprecise about the higher-order derivative claim. I recommend the authors to rephrase for correctness and better clarity.

---

> ### Author Response · Authors · 2020-11-21
> **Response to R4 Questions**
>
> Thank you for the detailed review and constructive feedback! Please find our responses below.
>
> 1) The teacher leaps towards the region explored by the student, and it starts from the initialization (i.e.,the current meta-model’s parameter stage). With the “leap” learning rate (see the $gamma$ hyper-parameter) and the steps (see $k$) executed by the student, the teacher presumably wouldn’t go as far as to a stage where it encounters the vanishing gradient problem.
>
> 2) Yes, we ignore the higher-order gradients in the same spirit as the first order MAML. The algorithm still solves the bilevel optimization problem by the gradients through the teacher’s “leap”. We acknowledge that both the “leap” and the student’s exploration contribute to the gradients, and both are informative. Ignoring the latter, however, does not hurt the quality of the former.
>
> 3) If we unroll the optimization iterations into a multi-layer neural network, the teacher’s leap essentially becomes a skip connection between different layers. Following the reviewer’s comment that the “gradients are passed through the skip connections in ResNet,” we pass the gradients through the teacher’s “leap”. We’ll improve the clarity of this part.
>
> 4) MAML is extremely expensive or even infeasible when exploring a long-horizon by the inner loop (e.g., $k>10$) because the computation graph would become too large to handle. In contrast, the size of our computation graph is constant to the inner-loop updates. This reasoning applies to computation, memory, and gradient vanishing or exploding. Please see Figure 3 in Appendix A.4 comparing our approach with MAML by varying the number of inner steps. We acknowledge that we have to limit the search space for $\alpha$ --- we use $b=1$ and $b=3$ in the experiments --- to speed up the inner loop.
>
> 5) Thank you for catching it, and we’ll make it more precise in the revised paper! We share the same argument made in the iMAML paper, “the meta-learning process requires higher-order derivatives, ...”, but we should be concrete by the “higher-order gradients”

---

### Official Review · AnonReviewer1 · 2020-11-05
**Interesting work but some claims seem exaggerated/confusing**

**Rating:** 4
**Confidence:** 4

**Review:**

This paper proposes a new method for gradient-based meta-learning that allows for multiple steps of inner-loop optimization while avoiding expensive backpropagation through these updates. The proposal is based on a teacher-student scheme, where the student does many steps of inner loop gradient descent, and the teacher does only a single update step where the single step is computed using the regions visited by the student. The authors propose variants of this setup for both Maml and Reptile. The meta-learning model is trained by back-propagating through the teacher's updates. Experiments are conducted on few-shot learning, long-tailed classification, and meta-attack benchmarks.

Pros
* The paper considers a variety of different benchmarks (other than simply few-shot learning), including memorization in few-shot learning, long-tailed classification and meta-attack.

Cons
* It is not straightforward to me what the benefit of the proposed method is compared to first-order maml (fomaml) or reptile. Both fomaml and reptile do not require expensive back-propogation through the inner-loop updates. It is mentioned in the paper that for their model, "one can update the meta-model by backpropagating gradients through the teacher’s “leap”, not the student’s update chain (ignoring that the chain starts from the meta-model)." This to me means that the second-order term is being ignored and so the update is indeed first-order.  Then, the only difference (as far as I can see), is that the proposed method uses the convex hull of the last b checkpoints and a linear combination that includes the initial parameters (which is a form of a lookahead optimizer) instead of simply the last checkpoint after k steps of gradient descent in the inner-loop. This could indeed offer some benefit but doesn't seem to offer the claimed benefit of "enabling...exploring long horizons in the inner loop" compared to fomaml or reptile as training with long inner loops seem similarly computationally expensive across all these first-order methods. If what I said above is not the case, I think the outer-gradient for the proposed method should be specifically stated and compared against the fomaml and second-order maml cases to highlight how this gradient is different and what benefit this offers (similar to the analysis that was done in the Reptile paper). In general, I believe a better argument needs to be made about the benefits of the proposed method compared to first-order methods such as fomaml and reptile, given that they all seem computationally similar in terms of cost.
* My interpretation is that their "lazy maml" is essentially inserting a lookahead optimizer in the inner loop while having a first-order outer update; while "lazy reptile" is inserting a lookahead optimized in the inner loop and the outer-loop is also a lookahead optimizer by definition of reptile. However, in addition to the lookahead optimizer, they add calculating the convex hull of the last b checkpoints and hyperparmeters for this calculation and the lookahead optimizer are computed during meta-training by doing grid search on the perfomance on the corresponding validation set in each episode. It is unclear what the value of this hyperparameter search is and since it complicates the implementation of their method, it would be useful to have experiments to quantify the necessity of this hyperparameter search.
* The experiments for TiredImageNet seem to only compare against second-order maml where the inner-loop consists of very few steps? I think a natural comparison is also fomaml where the number of inner-loop steps is the same as what was used for their model. Additionally, I believe results from other state-of-the-art models should also be shown to put the proposed model's results in context (a table is probably better for comparing across the metrics from different models). I had a similar observation about the few-shot memorization case - that fomaml with same number of inner-loop steps is a natural baseline. I would also be interested in knowing the metrics of iMAML for these benchmarks as it seems like a natural method for comparison.

To conclude, though some of the experimental results seem interesting, I believe the discussion of the method could be greatly improved as some of the claims seem a bit exaggerated or confusing. Additionally, I believe the setup of the experiments could be improved as stated above.

---

> ### Author Response · Authors · 2020-11-21
> **Response to R1 questions**
>
> Thank you for the detailed reviews and constructive feedback! Please see our responses to each of the “Cons” bullet points as follows.
>
> * We confirm the reviewer's comment on our work's connections to FOMAML, Reptile, and the lookahead optimizer. However, our approach outperforms FOMAML or Reptile by significant margins (see Tables 1, 5, and 6), which we believe attribute to three components. First, unlike lookahead, the teacher adapts the "leap" learning rate and explores multiple checkpoints visited by the student. These are made possible by the meta-learning problem setup. Second, unlike Reptile, the meta-learner still depends on the meta-gradients, which are arguably more reliable than the update rule in Reptile. Moreover, the dependency made it straightforward to apply our approach to the two-component weighting algorithm for long-tailed classification, but it's unclear how to employ Reptile in the algorithm. Third, unlike FOMAML's ignoring the second-order gradients, we stop much higher-order gradients and take advantage of the teacher's "leap," which is arguably more accurate than the single-step update in FOMAML.
>
> * The comparison between our approach and Reptile in Tables 1, 5, and 6 answers this question because, by slightly changing the reviewer's understanding of Reptile, we can view Reptile as a lookahead optimizer (for the inner loop) coupled with a special outer-loop update to the meta-model. The results below further answer this question by a comparison with FOMAML.
>
> * To save space for long-tailed classification and meta-attack, we use Tables 5 and 6 in Appendix A.6 to report additional results on few-shot learning. The iMAML numbers are from existing papers since we did not find any publicly available source code, and we could not reproduce a reliable implementation ourselves. Following the reviewer's suggestion, we perform the memorization experiment for FOMAML with the same number of steps as ours (k = 15). FOMAML yields 31.28 ± 0.42, which is better than MAML and TAML but worse than ours (38.37±0.40). We also test it on TieredImageNet for 20-way, 30-way, and 50-way few-shot learning. The differences between FOMAML and MAML diminish as the number of classes increases, but ours can maintain the margins, as shown below.
>
> #------------------------------------------------------------------------------------------------------#
>
> Method |                        20-way	|	            30-way |		         50-way  |
>
> MAML    |                    31.92 ± 0.4  |             25.95 0.33 |                 17.72 ± 0.2 |
>
> FO-MAML	|	     34.47 ± 0.4  |            26.80 ± 0.30 |              17.79 ± 0.2  |
>
> Ours (MAML)  |	     36.11 ± 0.42 |	        28.04 ± 0.34 | 	      19.70 ± 0.2  |
>
> #--------------------------------------------------------------------------------------------------------#

---

### Decision · Program_Chairs · 2021-01-07
**Final Decision**

**Decision:**

Reject

**Comment:**

This paper presents a variant of MAML or Reptile, where the meta-update along the long trajectory of the inner-loop optimization is bypassed to reduce the computational overhead appeared in MAML. The main idea is to use the look-ahed optimizer with careful tuning of relevant hyperparameters, which is done by a teacher-student scheme. Lazy MAML/Reptile are presented and experiments  demonstrated their validity. While the paper contains interesting ideas, most of reviewers have a few concerns which are not even resolved even after the author responses. First of all, ResNet analogy with respect to teacher update was claimed but it was never clearly shown in the paper. The method needs careful tuning of hyperparameters in the inner loop, but  the study about the computation requirements is not convincing yet.  Long inner loops are computationally feasible for both fomaml and reptile so its not clear in which way the proposed method is improving lengthy exploration in the inner loop other than the performance being better in the experiments. Improving the paper, taking these comments into account, will lead to a good work in near future.